# Clinical, Laboratory and Lung Ultrasound Assessment of Congestion in Patients with Acute Heart Failure

**DOI:** 10.3390/jcm11061642

**Published:** 2022-03-16

**Authors:** Alberto Palazzuoli, Isabella Evangelista, Matteo Beltrami, Filippo Pirrotta, Maria Cristina Tavera, Luigi Gennari, Gaetano Ruocco

**Affiliations:** 1Cardiovascular Diseases Unit, Cardio Thoracic and Vascular Department, Le Scotte Hospital, University of Siena, Viale Bracci 14, 53100 Siena, Italy; mctavera@gmail.com; 2Department of Internal Medicine, ASST Ovest Milanese, 20013 Magenta, Italy; isabella.evangelista87@gmail.com; 3Cardiology Unit, San Giovanni di Dio Hospital, 50143 Florence, Italy; beltrami.matteo1@gmail.com; 4Internal Medicine Unit, Department of Medical Sciences, University of Siena, 53100 Siena, Italy; pirrotta90@gmail.com (F.P.); gennari@unisi.it (L.G.); 5Cardiology Unit, Riuniti of Valdichiana Hospital, Usl Toscana SUD-EST, 53045 Montepulciano, Italy; gmruocco@virgilio.it

**Keywords:** heart failure, congestion, BNP, LUS

## Abstract

Congestion is the main cause of hospitalization in patients with acute heart failure (AHF), however its precise assessment by simple clinical evaluation remains elusive. The recent introduction of the lung ultrasound scan (LUS) allowed to physicians to more precisely quantify pulmonary congestion. The aim of this study was to compare clinical congestion (CC) with LUS and B-type natriuretic peptide (BNP) in order to achieve a more complete evaluation and to evaluate the prognostic power of each measurement. Methods: All patients were submitted to clinical evaluation for blood sample analysis and LUS at admission and before discharge. LUS protocol evaluated the number of B-lines for each chest zone by standardized eight site protocol. CC was measured following ESC criteria. The mean difference between admission and discharge congestion logBNP and B-lines values were calculated. Combined end points of death and rehospitalization was calculated over 180 days. Results: 213 patients were included in the protocol; 133 experienced heart failure with reduced ejection fraction (HFrEF), and 83 presented with heart failure with preserved ejection fraction (HFpEF). Patients with HFrEF had a more increased level of BNP (1150 (812–1790) vs. 851 (694–1196); *p* = 0.002) and B lines total number (32 (27–38) vs. 30 (25–36); *p* = 0.05). A positive correlation was found between log BNP and Blines number in both HFrEF (r = 0.57; *p* < 0.001) and HFpEF (r = 0.36; *p* = 0.001). Similarly, dividing B-lines among tertiles the upper group (B-lines ≥ 36) had an increased clinical congestion score. Among three variables at admission only B-lines were predictive for outcome (AUC 0.68 *p* < 0.001) but not LogBNP and CC score. During 180 days of follow-up, univariate analysis showed that persistent ΔB-lines <−32.3% (HR 6.54 (4.19–10.20); *p* < 0.001), persistent ΔBNP < −43.8% (HR 2.48 (1.69–3.63); *p* < 0.001) and persistent ΔCC < 50% (HR 4.25 (2.90–6.21); *p* < 0.001) were all significantly related to adverse outcome. Multivariable analysis confirmed that persistent ΔB-lines (HR 4.38 (2.64–7.29); *p* < 0.001), ΔBNP (HR 1.74 (1.11–2.74); *p* = 0.016) and ΔCC (HR 3.38 (2.10–5.44); *p* < 0.001 were associated with the combined end point. Conclusions: a complete clinical laboratory and LUS assessment better recognized different congestion occurrence in AHF. The difference between admission and discharge B-lines provides useful prognostic information compared to traditional clinical evaluation.

## 1. Introduction

Congestion is the main cause of hospital admission in patients with acute decompensated heart failure (ADHF). ADHF recurrence is often the consequence of fluid retention, which leads to both systemic and pulmonary congestion [1,2,3]. Moreover, the lack of congestion resolution during HF hospitalization should be related to poor prognosis. In this scenario, a multiparametric congestion evaluation in order to better stratify ADHF patients at high risk for re-hospitalization would be useful [4,5]. Despite the residual clinical congestion after discharge remains a strong predictor of worse outcomes, any congestion scores are not sensitive enough to detect the real cardiac filling pressure, and they do not precisely reflect the real intravascular and extravascular fluid overload. Traditionally, the most used strategy is the clinical evaluation, which is associated with chest radiography (CRx) and natriuretic peptides (NPs) measurement [6,7]. However, this approach is not readily available in a primary care setting and at the patient’s home. In this context, a precise and non-invasive evaluation of congestion by clinical, ultrasound, and laboratory tools appears to be mandatory, although up to now, there does not exist a precise algorithm evaluating systemic and pulmonary congestion all at once. Therefore, systemic congestion is not strictly related to cardiac congestion, and the simple absence of clinical signs of fluid overload cannot exclude an increased left ventricular filling pressure (LVEDP) [8]. Although natriuretic peptides (NP) are included in the diagnostic HF algorithm, recent data are cause for doubt about their prognostic significance, and NP guided therapy has not demonstrated significant benefit compared with the standard approach which is based on clinical signs [9,10]. Interestingly, lung ultrasound (LUS) examination has been proposed as a simple, available, and economic tool to recognize pulmonary congestion in ADHF. LUS is an integrated non-invasive tool based on the recognition of artefact reverberations arising from extravascular lung water and inter-lobar septal imbibition due to lung fluid accumulation. The number of ultrasound lung comets (B-lines) is directly related to the lung congestion degree [11,12]. Recently, some reports demonstrated the prognostic power of B-lines counts in acute HF, as well as the importance of this tool for guiding to a tailored treatment to achieve adequate decongestion [13,14,15]. Therefore, LUS analysis could be accounted for together with clinical evaluation and B-type natriuretic peptide (BNP) measurement in order to better identify the congestion status and degree during hospitalization in patients affected by ADHF. Accordingly, we aim to evaluate the congestion at admission and at discharge through three different methods as clinical examination, BNP measurement, and LUS assessment in heart failure with reduced or preserved ejection fraction; we assess the prognostic impact of different congestion evaluation during a mean follow-up of 180 days after discharge.

## 2. Methods

A prospective cohort of 248 patients admitted with a main diagnosis of ADHF were enrolled within 12 h of hospital admission. Consecutive patients were admitted to the Cardiovascular Diseases Unit of the Le Scotte Hospital in Siena. The inclusion criteria were the presence of signs and/or symptoms of AHF, irrespective of the aetiology, associated with chest radiography suggestive for HF. Exclusion criteria were (1) patients with a poor acoustic chest and cardiac window; (2) pulmonary disease as the main cause of dyspnoea (e.g., pulmonary asthma, chronic obstructive pulmonary disease interstitial lung disease); (3) a history of pneumothorax, lobectomy or lung cancer; (4) ST elevation myocardial infarction within the last two months; (5) patients with unstable cardiogenic shock (blood pressure < 90 mmHg); (6) patients with infection, inflammatory, autoimmune or neoplastic diseases. 

## 3. Physical Examination and Blood Tests

Patients were evaluated by two physicians at admission and discharge to assess the grade of clinical congestion (CC) score, giving 1 point for each of following signs: pulmonary rales, third heart sound, jugular venous distention, peripheral edema and hepatomegaly (5 total points) [13].

Blood tests were performed, and hemocrome, electrolytes, renal function glycemia, uricemia, plasma osmolarity, troponin, and B-type natriuretic peptide were measured. The BNP assay was evaluated at admission and before discharge by Biosite Inc., San Diego, CA, USA. 

## 4. Echocardiography

Left ventricular ejection fraction (LVEF) was measured using the biplane Simpson’s method. Diastolic function was assessed from the pattern of mitral inflow by pulsed-wave Doppler. Mitral annular early diastolic velocity (e’) was assessed at the septal and lateral sites of the mitral annulus using tissue Doppler imaging. E/A ratio, e’ wave peak velocity, E/e’ were calculated. Inferior vena cava (IVC) and tricuspid annular plane systolic excursion (TAPSE) were obtained as recommended. Pulmonary arterial systolic pressure (PASP) was estimated using tricuspid rigurgitation velocity, and ICV diameter and collapse which were analyzed by epigastric scan according to recent guidelines [16]. Patients were classified according to LVEF, reduced EF (LVEF < 50%) and preserved EF (LVEF ≥ 50%).

## 5. Lung Ultrasound

Measurements were averaged for each LUS zone. Eight LUS zones were analysed for each patient. LUS was performed, placing patients in a semi-supine position (45°), using the same probe used for echocardiography. We evaluated eight chest zones [12], two parasternal chest scans and two scans of the anterior and lateral basal chest was obtained on the right and left hemi thoraxes. LUS was performed twice in all patients: the first evaluation was done at admission at the end of standard echocardiography. The maximum number of B-lines was counted in each single scan and the sum of all eight zones was calculated as total number of B-lines counts. The second evaluation was performed before discharge. Inter- and intra-observer variability ranges from 5 to 8%. 

### 5.1. Follow-Up

The composite primary outcome was considered as death for cardiovascular causes and hospitalisations primarily due to heart failure within 180 days after discharge. All patients were followed by direct clinical check-up or contacted by phone or remotely for six months after discharge and were reviewed by different clinicians who provided the local HF service. Clinical follow-up was obtained through information from subsequent patient visits or by telephone or internet contacts. 

### 5.2. Statistical Analysis

Continuous variables were expressed as median and interquartile range (IQR) and categorical variables as count or percentage. Differences for patients with HFrEF compared to HFpEF were tested using a Mann-Whitney non-parametric test and X^2^ tests. Spearman’s rho correlation coefficient was used to assess relationships for continuous variables; BNP was analysed after logarithmic transformation. A receiving operating characteristic (ROC) curve was used to assess the relationship between variables and outcomes. Mean differences between admission and discharge for CC score BNP and B-lines were calculated. Cox regression analysis was used to assess the independent and the dependent relationship between variables (B-lines; BNP; congestion score; age; sex; cardiovascular risk factors, including hypertension, diabetes, dyslipidaemia, smoking, and coronary artery disease) and 180 days outcome. All reported probability values were two-tailed, and a *p* value ≤ 0.05 was considered statistically significant. Statistical analysis was performed using the SPSS 20.0 statistical software package (SPSS Inc., Chicago, IL, USA).

## 6. Results

### Baseline Characteristics

Of 216 patients enrolled, 133 experienced heart failure with reduced ejection fraction (HFrEF), and 83 presented heart failure with preserved ejection fraction (HFpEF). Compared to patients with HFpEF, patients with HFrEF were more likely to be male (54% vs. 36%; *p* = 0.01), have lower BMI values (27.4 (24.5–29.7) vs. 28.7 (26.0–30.5) kg/m^2^; *p* = 0.02) and to have higher BNP levels at baseline (1150 (812–1790) vs. 851 (694–1196); *p* = 0.002). Patients with HFrEF experienced a higher percentage of CAD (66% vs. 23%; *p* < 0.001) and diabetes (50% vs. 32%; *p* = 0.01), and a lower prevalence of hypertension (81% vs. 51%; *p* < 0.001). No difference was observed regarding the incidence of dyslipidaemias, AF, smoke, and signs of congestion between HFrEF and HFpEF patients. 

Compared to patients with HFpEF, at echocardiographic assessment the HFrEF population revealed higher LVEDVi (160 (140–190) vs. 115 (100–145) mL/m^2^; *p* < 0.001), LVESVi (100 (80–130) vs. 55 (45–70) mL/m^2^; *p* < 0.001), LA area (28 (24–31) vs. 25 (22–27) cm^2^; *p* < 0.001) and lower TAPSE (18 (16–21) vs. 20 (17–22) mm; *p* = 0.02). Overall, adverse events occurred in 53 patients; no differences were observed in terms of all cause mortality and HF hospitalization between HFrEF and HFpEF (Table 1).

B-lines at hospital admission were slightly increased in HFrEF vs. HFpEF without statistical significance (32 (27–38) vs. 30 (25–36); *p* = 0.07). We divided our sample into three subgroups (1° tertile: B-lines ≤ 27, *n* = 63; 2° tertile: B-lines between 28 and 35, *n* = 83; 3° tertile: B-lines ≥ 36, *n* = 70). Statistical analyses highlighted a prominent clinical congestion in patients with B-lines ≥ 36 compared to other tertiles in terms of jugular vein distention (17% vs. 25% vs. 43%; *p* = 0.004), hepatomegaly (21% vs. 37% vs. 43%; *p* = 0.02), third heart sound (22% vs. 23% vs. 43%; *p* = 0.009), and BNP (822 (586–1130) vs. 890 (694–1354) vs. 1740 (982–2577)) (Table 2).

Overall, at admission there was a positive correlation among logBNP and B-lines both in HFrEF (r = 0.57; *p* < 0.001) and HFpEF (r = 0.36; *p* = 0.001) and at discharge among logBNP and B-lines in HFrEF (r = 0.73; *p* < 0.001) and HFpEF patients (r = 0.53; *p* < 0.001) (Figure 1).

## 7. Predictors of Outcome at Admission

Univariate and multivariate logistic regression analyses were performed to evaluate the prognostic role of congestion parameter at hospital admission and discharge. At admission, on univariate analyses, CC score > 3 at admission (HR 8.20 (4.74–14.16); *p* < 0.001), B-lines (HR 1.07 (1.03–1.11); *p* < 0.001), peripheral oedema (HR 3.44 (1.77–6.68); *p* < 0.001), hepatomegaly (HR 3.18 (1.84–5.50); *p* < 0.001), jugular vein distention (HR 3.48 (2.03–5.98); *p* < 0.001) and third heart sound (HR 2.66 (1.55–4.56); *p* < 0.001) were related to all cause mortality and HF hospitalization at 60 days follow-up. Conversely, logBNP (HR 1.47 (0.95–2.29), *p* = 0.08) did not correlate with outcome. Multivariate analyses demonstrated congestion score > 3 was an independent predictor of all-cause mortality and HF hospitalization and after adjustment for logBNP (HR 9.83 (5.27–18.31); *p* < 0.001), B-lines (HR 6.81 (3.82–12.13); *p* < 0.001) and for logBNP and B-lines (HR 8.26 (4.46–15.26); *p* < 0.001). Conversely, Congestion score > 2 was not related to outcome also after adjustment for logBNP and B-lines (Table 3).

An ROC curve showed the relationship between the primary outcome and B-lines at admission (AUC: 0.68 (0.60–0.77); *p* < 0.001); conversely, logBNP at admission was not associated to the composite outcome (AUC: 0.57 (0.47–0.66); *p* = 0.15) (Figure 2).

## 8. Congestion Differences between Admission and Discharge

We calculate the mean difference between admission and discharge congestion logBNP and B-lines values and we analysed these cutoffs in relation to outcome. Based on our analysis and median values, we considered a CC reduction < 50%, ΔBNP < −43.8% and ΔB-lines < −32.3% as significant improvement. Conversely, subjects with mean clinical laboratory and LUS values above the mean reduction were defined as the persistent group. Mortality at 60 days occurs in 69% of subjects with persistent congestion, vs. 8% of patients with improved congestion. (*p* < 0.0001); 37% in those with unsolved ΔBNP vs. 11% in those with more significant BNP decrease (*p* < 0.001); 44% of patients with persistent B-lines vs. 5% of patients with significant improvement (*p* < 0.0001). Relative risk calculation demonstrates an excellent performance for the all Δ at both 60 and 180 days: persistent ΔCC RR 7.7 (IC 4.1–14.5) at 60 days and RR 2.0 at 180 days (IC 1.6–2.6 *p* < 0.001); persistent ΔBNP RR 3.4 (IC 1.9–6.1) at 60 days and RR 1.7 (IC 1.3–2.2) at 180 days (*p* < 0.001); persistent ΔB-lines RR 9.6 (IC 3.9–23) at 60 days and RR 3.3 (IC 2.3–4.7) *p* < 0.001 at 180 days compared to the resolved group.

During 180 days of follow-up, univariate analysis showed that persistent ΔB-lines (HR 6.54 (4.19–10.20); *p* < 0.001), persistent ΔBNP (HR 2.48 (1.69–3.63); *p* < 0.001) and persistent ΔCC (HR 4.25 (2.90–6.21); *p* < 0.001) were all significantly related to adverse outcome. Multivariable analysis confirmed that persistent ΔB-lines (HR 4.38 (2.64–7.29); *p* < 0.001), persistent ΔBNP (HR 1.74 (1.11–2.74); *p* = 0.016) and persistent ΔCC (HR 3.38 (2.10–5.44); *p* < 0.001) were associated with composite outcome (Table 4).

A Kaplan Meier survival curve confirmed the additional role of Δ congestion and ΔB-Lines in risk stratification (Figure 3).

## 9. Discussion

The current analysis confirmed the prognostic role of congestion evaluation at admission and discharge, however the combined analysis including B-lines count and BNP measurement is much more accurate for outcome prediction. More specifically, a high CC score at admission and discharge as evaluated by the application of the Gheorghiade scale demonstrated a good accuracy, confirming that clinical assessment remains an important feature during acute HF management [5]. In contrast, BNP measurement at admission was not related with outcome, confirming some results of the GUIDE trial [16]. Indeed, BNP is prone to several forms of bias not strictly related to congestion: intrinsic cardiac hemodynamic conditions such as wall stretching, associated mitral valve disease, atrial fibrillation, and right ventricular dysfunction may influence laboratory values. Therefore, systemic conditions such as high BMI, the presence of CKD, sarcopenia and inflammatory status are further biases. The serial LUS evaluation confirmed its diagnostic and prognostic role: subjects with a poor decrease in B-lines numbers from admission to discharge are much more prone to develop adverse events; conversely, significant reduction of B-lines during hospitalization is associated with good outcomes [17]. These findings highlight two factors: a significant percentage of patients hospitalized for Acute HF do not reach an optimal decongestion at discharge; and secondly, B-lines assessment and repeated LUS evaluation is an important approach for pulmonary congestion monitoring across the hospitalization period, and it could be inserted as ab additional tool during AHF management. Many hospitalized patients are usually discharged after symptom recovery without precise decongestion evaluation just to reduce hospitalization length and health costs. However, this approach demonstrated several weaknesses in terms of early hospital readmission and treatment efficacy [18]. Our findings are in line with previous studies evaluating the prognostic relevance of clinical congestion, and they appear to be integrative, suggesting that a residual congestion evaluated with different methods is related with increased risk [19,20]. Overall, the multiparametric diagnostic congestion approach combining clinical score with BNP assay and LUS evaluation showed a better accuracy compared with each method singly applied, and it is associated with a more accurate risk stratification [21]. Additionally, current policy may avoid invasive monitoring and become integrative for the congestion evaluation in different sites. It is well known that the best strategy for congestion evaluation remains right heart catheterization with direct measurement of wedge and right atrial pressures, although the invasive method limits any extensive application. Furthermore, the ESCAPE trial did not demonstrate any significant benefit in invasive monitoring assessment with respect with a clinical evaluation [22]. This is probably due to different mechanisms contributing to cardiac and systemic congestion: the former is closely related with cardiac invasive measurement, whereas the latter are due to hydrostatic and oncotic pressure, interstitial tissue condition, and Na tubular resorption directly related to systemic water retention. Interestingly, serial and repetitive screening during hospitalization is helpful for reducing hospitalization and congestion resolution identification [8]. Accordingly, in ambulatory patients with advanced HFrEF, the continued remote monitorization of pulmonary pressure by a Cardio-Mems system demonstrated a significant reduction in HF related hospitalization by a customized therapy [23]. Previous reports confirmed the importance of LUS assessment at admission and discharge: in a single center study, Gargani et al., showed discharge B-lines numbering < 15 were associated with reduced risk of HF hospitalization [24]. Similarly, Coiro revealed that a residual B-line > 30 independently predicts outcome, and an algorithm including LUS with BNP and NYHA class identifies patients with higher risk for hospitalization recurrence [25]. In a larger sample size of patients with both HF with reduced or preserved EF, Palazzuoli et al. revealed a discharge cut-off >22 b-lines for HFrEF and 18 for HFpEF were predictive for adverse outcome [26]. In a simplified model assessing LUS in four chest zones, Platz et al. stratified patients by mean B-lines reduction over six days from hospital admission [9]. Similarly, same authors in a retrospective study revealed that in both acute and chronic conditions, B-lines numbers change rapidly, and an LUS exam is an appropriate strategy for treatment monitoring and response [27]. Finally, our findings are in accordance with a recent position paper suggesting that a multiple ultrasound scan evaluation including imaging of the heart, lung, venous system and kidney is the optimal approach for cardiac pulmonary and systemic congestion evaluation, respectively [28]. 

## 10. Study Limitations

This is a single centre study with a relatively small sample size, but this is one of the larger studies with contemporary clinical, biochemical and LUS evaluation in patients hospitalised for acute HF. This is a post-hoc analysis, despite the fact that the patients were prospectively recruited and only patients with complete clinical, echocardiographic and follow-up data were included. 

We used a simplified eight-zone protocol which is more practical for clinical use in the emergency department, but other studies applied a different scan protocol ranging from 4 to 28 chest zones. Current differences may partially explain the different B-lines counts compared with previous analyses. Other conditions, such as interstitial lung diseases, acute respiratory distress syndrome and interstitial pneumonia can confound B-lines analysis. Patient selection were performed by chest radiography and BNP level associated with dyspnoea in accordance with ESC guidelines, but some respiratory or extra cardiac conditions may have influenced the enrolment criteria. We did not divide our patients according to the ESC classification in HFrEF, HFpEF, and heart failure with mild reduced EF because only few patients (*n*. 32) had EF ranging from 41 to 49%, thus we included this group into HFrEF. In addition, physicians conducting clinical examinations and echocardiography were aware of the CRx and BNP results. This might have introduced further bias. Our results may partially depend on decongestion treatment, intensity of care, and diuretic amount adopted before and during hospitalization. Finally, the persistence of pre-discharge B-lines may imply sub-optimal decongestion treatment. However, we cannot assert that a more elevated diuretic dose could drive towards relevant B-lines reduction and better outcomes. Therefore, a multi-centre analysis comparing a larger sample size LUS with various congestion scores is welcome and might support or refute our findings. 

## 11. Conclusions

Our data confirmed the importance of achieving congestion status resolution before discharge by an integrated clinical laboratory and LUS evaluation. The reduction of the B-lines number from admission to discharge appears to be the most powerful predictor for HF recurrence and death in patients hospitalized for HF. Future studies with larger sample would shed light on the efficacy of the current strategy in optimizing time discharge and in reducing adverse events.

## Figures and Tables

**Figure 1 jcm-11-01642-f001:**
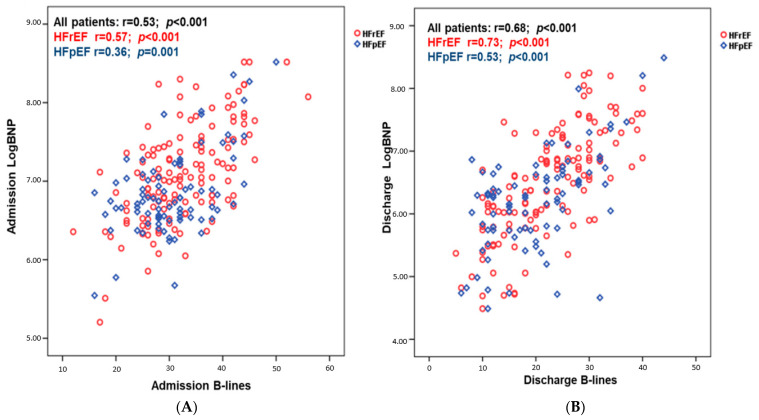
Correlation existing between LogBNP and B-lines number at admission (**A**) and at discharge (**B**) dividing patients according to ejection fraction.

**Figure 2 jcm-11-01642-f002:**
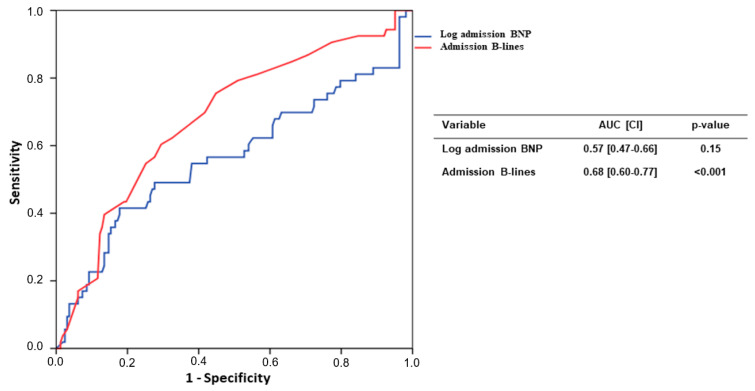
ROC curve analysis of admission BNP values and B-lines number.

**Figure 3 jcm-11-01642-f003:**
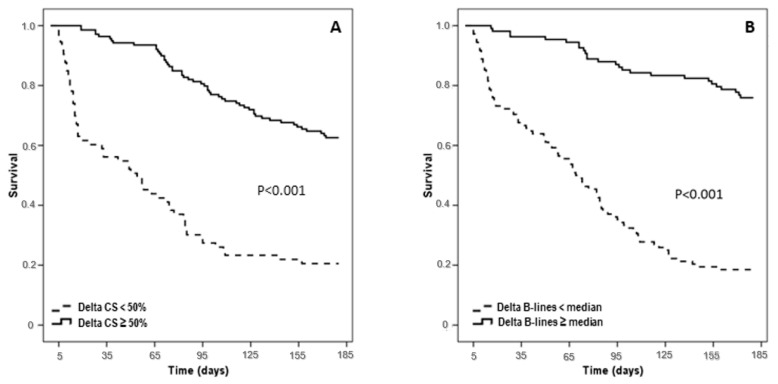
Kaplan-Meyer curve of median Δ Congestion score (CS) (**A**) and Δ B-lines (**B**) during follow-up (180 days).

**Table 1 jcm-11-01642-t001:** Clinical risk factors and echocardiographic features of enrolled patients divided by ejection fraction (EF).

Characteristic	HFrEF *n* = 133	HFpEF *n* = 83	*p*-Value
Age (years)	82 (77–87)	79 (77–83)	0.04
Men—*n*. (%)	72 (54)	30 (36)	0.01
BMI (kg/m^2^)	27.4 (24.5–29.7)	28.7 (26.0–30.5)	0.02
Risk factors—*n*. (%)			
CAD	88 (66)	19 (23)	<0.001
Diabetes	66 (50)	27 (32)	0.01
Dyslipidaemia	69 (52)	37 (46)	0.30
Hypertension	68 (51)	67 (81)	<0.001
Smoking	38 (29)	29 (35)	0.32
AF	38 (29)	16 (19)	0.12
LVEF	33 (25–44)	56 (50–62)	<0.01
Clinical examination & BNP			
Heart rate (beats/min)	90 (87–97)	89 (86–94)	0.36
Systolic blood pressure (mmHg)	125 (110–135)	140 (130–150)	<0.001
Diastolic blood pressure (mmHg)	70 (55–80)	85 (80–95)	<0.001
Respiratory rate (*n*./min)	29 (27–32)	29 (26–33)	0.58
Rales *n*./(%)	103 (77)	72 (86)	0.09
Peripheral oedema *n*./(%)	78 (58)	44 (53)	0.42
Hepatomegaly *n*./(%)	50 (38)	24 (29)	0.19
Jugular vein distention *n*./(%)	38 (28)	24 (29)	0.95
Third heart sound *n*./(%)	42 (31)	21 (25)	0.32
BNP (pg/mL)	1150 (812–1790)	851 (694–1196)	0.002
Echocardiography and LUS			
LVEDD (mm)	61 (56–66)	51 (45–55)	<0.001
LVESD (mm)	46 (41–52)	34 (29–37)	<0.001
LVEDVi (mL/min²)	160 (140–190)	115 (100–145)	<0.001
LVESVi (mL/min²)	100 (80–130)	55 (45–70)	<0.001
Left atrial Area (cm^2^)	28 (24–31)	25 (22–27)	<0.001
PASP (mmHg)	45 (40–50)	45 (40–55)	0.92
Septal thickness (mm)	11 (10–13)	12 (11–14)	0.001
Posterior wall (mm)	11 (9–12)	12 (11–13)	0.002
TAPSE (mm)	18 (16–21)	20 (17–22)	0.02
Inferior cave vein diameter (mm)	23 (22–24)	22 (21–25)	0.95
E/e’	16 (14–18)	16 (14–18)	0.63
B-lines (*n*)	32 (27–38)	30 (25–36)	0.07
Outcome			
60 days adverse events—*n*. (%)	36 (27)	17 (21)	0.27

Abbreviations: Atrial Fibrillation (AF); Bpdy Mass Index (BMI); B-type Natriuretic Peptide (BNP); Coronary Artery Disease (CAD); Heart Failure with preserved ejection fraction (HFpEF); Heart Failure with reduced ejection fraction (HFrEF); Left Ventricular End-Diastolic Diameter (LVEDD); Left Ventricular End-Diastolic Volume indexed (LVEDVi); Left Ventricular Ejection Fraction (LVEF); Left Ventricular End-Systolic Diameter (LVESD); Left Ventricular End-Systolic Volume indexed (LVESVi); Lung Ultrasound (LUS); Pulmonary Artery Systolic Pressure (PASP); Tricuspid Anular Plane Systolic Excursion (TAPSE).

**Table 2 jcm-11-01642-t002:** Clinical, biochemical and ultrasound prevalence of congestion according to tertiles of B-lines.

	B-Lines Tertile 1 (Range: ≤27) *n* = 63	B-Lines Tertile 2 (Range: 28–35) *n* = 83	B-Lines Tertile 3 (Range: ≥36) *n* = 70	*p*-Value
Clinical congestion				
Rales (yes)—*n*. (%)	51 (81)	64 (77)	60 (86)	0.40
Peripheral oedema—*n*. (%)	31 (49)	47 (57)	44 (63)	0.28
JV distention—*n*. (%)	11 (17)	21 (25)	30 (43)	0.004
Hepatomegaly—*n*. (%)	13 (21)	31 (37)	30 (43)	0.02
Third heart sound—*n*. (%)	14 (22)	19 (23)	30 (43)	0.009
Biochemical or ultrasound congestion				
BNP—pg/mL	822 (586–1130)	890 (694–1354)	1740 (982–2577)	<0.001
BNP—pg/mL (if in SR)	836 (672–1131)	974 (759–1383)	1525 (915–2595)	<0.001
BNP—pg/mL (if in AF)	586 (408–1110)	681 (473–815)	1900 (1410–2572)	<0.001
ICV—mm	22 (21–24)	22 (20–25)	24 (22–26)	0.002

Abbreviations: B-type natriuretic peptide (BNP); Inferior cave vein (ICV); Jugular vein (JV).

**Table 3 jcm-11-01642-t003:** Univariate and Multivariable analysis for admission clinical congestion, B-lines, and BNP including three combined models.

Variables	Association with the Composite of First HFH or Death Multivariable Analysis							
	Univariate		Model including LogBNP		Model Including B-Lines		Model Including LogBNP and B-Lines	
	HR (95% CI)	*p*-Value	HR (95% CI)	*p*-Value	HR (95% CI)	*p*-Value	HR (95% CI)	*p*-Value
Admission Congestion score > 3	8.20 (4.74–14.16)	<0.001	9.83 (5.27–18.31)	<0.001	6.81 (3.82–12.13)	<0.001	8.26 (4.46–15.26)	<0.001
Admission Congestion score ≥ 2	2.11 (0.95–4.67)	0.07	2.03 (0.91–4.50)	0.08	1.81 (0.81–4.04)	0.14	1.81 (0.81–4.03)	0.15
LogBNP	1.47 (0.95–2.29)	0.08	/	/	/	/	/	/
B-lines	1.07 (1.03–1.10)	<0.001	/	/	/	/	/	/

**Table 4 jcm-11-01642-t004:** Univariate and multivariable analysis for 60 days and 180 days outcome regarding the mean Δ values of clinical congestion (CC) score, Log BNP and B-lines calculated as the differences between admission and discharge.

Variables	60 Days				180 Days			
	Univariate HR (CI)	*p*-Value	Multivariable HR (CI) *	*p*-Value	Univariate HR (CI)	*p*-Value	Multivariable HR (CI) *	*p*-Value
Persistent ΔB-lines (<−32.3%)	12.36 (4.92–31.07)	<0.001	7.52 (2.16–26.21)	0.002	6.54 (4.19–10.20)	<0.001	4.38 (2.64–7.29)	<0.001
Persistent ΔBNP (<−43.8%)	4.26 (2.23–8.10)	<0.001	1.54 (0.69–3.41)	0.29	2.48 (1.69–3.63)	<0.001	1.74 (1.11–2.74)	0.016
Persistent ΔCC (<50%)	12.13 (5.87–25.06)	<0.001	11.64 (4.65–29.10)	<0.001	4.25 (2.90–6.21)	<0.001	3.38 (2.10–5.44)	<0.001

* Adjusted for age, gender, smoking, hypertension, diabetes mellitus, dyslipidaemia, CAD, LVEF < 50%, AF.

## Data Availability

Data are available from Dr. Alberto Palazzuoli after official request.

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
