# Peer review of "Clinical, Laboratory and Lung Ultrasound Assessment of Congestion in Patients with Acute Heart Failure"

_jcm, 2022, doi:10.3390/jcm11061642_

Round 1

Reviewer 1 Report

JCM review, article 1625993

In this article Palazzuoli A. and colleagues have investigated the lung ultrasound scan (LUS) as a tool for quantification of pulmonary congestion. They found that LUS is an accurate tool, comparable with clinical congestion evaluation and BNP levels. LUS was a strong predictor for mortality and HF hospitalization, with better diagnostic performance than clinical evaluation and BNP.

This trial is important as it further expands the current body of evidence regarding the utility of LUS in quantification of pulmonary congestion. This tool is especially important at discharge, where it may identify those patients with persistent congestion, therefore at risk for worse prognosis.  

I have some comments which would help to improve this article:

The cohort was divided into HFpEF and HFrEF groups, but in the methods section, it is stated that patients were classified to three groups, including HFmrEF. First, according to the 2021 ESC guidelines this group is termed "mildly reduced" and not "mid-range" and it includes only LVEF of 41 to 49%. Second, was this group included in the trial? If it was, where? The author should clarify this. The author should also add on Table 1 the median LVEF for each group.

The timeframe of the primary outcome (composite of all-cause death and HF hospitalization) is inconsistently reciprocating between 60 and 180 days. Moreover, the definition of the primary outcome is not included in the summary. The author should decide what is the time frame for the outcome – be it 60 or 180 days or both - and specify it throughout the manuscript, including in the summary. For reasons of clarity and conciseness it is advised to choose either 60 or 180 days, but not both. Also, the author should clarify at the text whether "adverse events" relate to the composite outcome.

The "congestion differences between admission and discharge" sub-section of the results – the author should name those groups of patients who had less than median improvement in congestion, BNP and B-lines - persistent congestion, non-resolved, non-improves ect. The author should use this name for the groups in Table 4 and in the summary.  This would make the text more readable and comprehensible.

The author should include at the discussion a reference to a more modern minimally- invasive evaluation of hemodynamic parameters – devices like cardioMEMS.  

Conformity: The term "B-lines" is spelled in 4 different ways throughout the manuscript (e.g. B lines, Blines and B-lines in three consecutive lines at the summary). The author should stick with only one spelling.

Punctuation: The author should use period (and only one!) at the end of every sentence, add one space after every period or comma, omit unnecessary spaces, use period and not comma for decimal point. 

Grammar: The author should correct the manuscript in term of grammar throughout the manuscript. Here are some examples:

Add "s" for present simple verbs in third person singular.

At the methods section - correct "blood test" to "blood tests".

At the first sentence of the discussion – add "and" before "at discharge"

One final comment: In your discussion, you conclude that the integrated diagnostic approach is superior to each individual one. I am not sure that this conclusion is indeed valid, based on your study deign and results , since you actually compared the diagnostic methods (one vs. another). Since the authors did not prove superiority of the integrated approach using LUS, it is at best a hypothesis generating inference.

Author Response

Reviewer 1

In this article Palazzuoli A. and colleagues have investigated the lung ultrasound scan (LUS) as a tool for quantification of pulmonary congestion. They found that LUS is an accurate tool, comparable with clinical congestion evaluation and BNP levels. LUS was a strong predictor for mortality and HF hospitalization, with better diagnostic performance than clinical evaluation and BNP.

This trial is important as it further expands the current body of evidence regarding the utility of LUS in quantification of pulmonary congestion. This tool is especially important at discharge, where it may identify those patients with persistent congestion, therefore at risk for worse prognosis. 

We  would thank this reviewer for its appreciation and for the insightful comments

I have some comments which would help to improve this article:

The cohort was divided into HFpEF and HFrEF groups, but in the methods section, it is stated that patients were classified to three groups, including HFmrEF. First, according to the 2021 ESC guidelines this group is termed "mildly reduced" and not "mid-range" and it includes only LVEF of 41 to 49%. Second, was this group included in the trial? If it was, where? The author should clarify this. The author should also add on Table 1 the median LVEF for each group.

Many thanks for this suggestion. We divided our sample between two groups based on EF threshold because the mid range group ( 49-40%) was restricted to small number. Of note we change our methods accordingly. We reported this statemen in Limitation as follows” We did not divide our patients according to the ESC classification in HFrEF, HFpEF and heart failure with mild reduced EF because only few patients (n.32) had EF ranging between 40-49%, thus we included this group into HFrEF”.

We added the mean EF for each group into table 1

The timeframe of the primary outcome (composite of all-cause death and HF hospitalization) is inconsistently reciprocating between 60 and 180 days. Moreover, the definition of the primary outcome is not included in the summary. The author should decide what is the time frame for the outcome – be it 60 or 180 days or both - and specify it throughout the manuscript, including in the summary. For reasons of clarity and conciseness it is advised to choose either 60 or 180 days, but not both. Also, the author should clarify at the text whether "adverse events" relate to the composite outcome.

We thank the reviewer for this clarification. We would clarify that combined end point was evaluated at both 60 and 180 days. For readers and methods simplification, we reported the most significant findings at 180 days. Thus, we modified the text accordingly including this sentence into Summary and Methods.

 The "congestion differences between admission and discharge" sub-section of the results – the author should name those groups of patients who had less than median improvement in congestion, BNP and B-lines - persistent congestion, non-resolved, non-improves ect. The author should use this name for the groups in Table 4 and in the summary.  This would make the text more readable and comprehensible.

 We are grateful for this comment, and we modified Results section to clarify our findings as follows” Based on our analysis and median values we considered a CC reduction< 50%, ΔBNP<-43.8% and ΔB-lines < -32.3% defined as significant   improvement. Conversely, subjects with mean clinical laboratory and LUS value above the mean reduction were defined as persistent group” subsequently we re- named the groups with new definition into the text and table 4

The author should include at the discussion a reference to a more modern minimally- invasive evaluation of hemodynamic parameters – devices like cardioMEMS. 

Accordingly in ambulatory patients with advanced heart failure the remote monitorization of pulmonary pressure by Cardio-Mems system demonstrated a significant reduction in HF related hospitalization by a customized therapy. We added the CHAMPION trial reference ( n.24)

Conformity: The term "B-lines" is spelled in 4 different ways throughout the manuscript (e.g. B lines, Blines and B-lines in three consecutive lines at the summary). The author should stick with only one spelling.

Punctuation: The author should use period (and only one!) at the end of every sentence, add one space after every period or comma, omit unnecessary spaces, use period and not comma for decimal point.

We apologize for these mistakes, and we modified punctuation in right way

Grammar: The author should correct the manuscript in term of grammar throughout the manuscript. Here are some examples:

Our apologizes for grammar defects and we correct them

Add "s" for present simple verbs in third person singular.

At the methods section - correct "blood test" to "blood tests".

At the first sentence of the discussion – add "and" before "at discharge"

done

One final comment: In your discussion, you conclude that the integrated diagnostic approach is superior to each individual one. I am not sure that this conclusion is indeed valid, based on your study deign and results, since you actually compared the diagnostic methods (one vs. another). Since the authors did not prove superiority of the integrated approach using LUS, it is at best a hypothesis generating inference.

 We agree we have not executed a specific step wise analysis for each method to assess the additional prognostic power of combined strategy, however our findings highlight the importance of multiple evaluations for screening congestion and improve risk stratification. Accordingly, we modified the final sentence in Summary as follows” Differences between admission and discharge B-lines provide useful prognostic information compared to the traditional clinical evaluation”.

 We would also clarify that in the Discussion section we cited the Pellicori document ( ref n.21)suggesting the combined ultrasound and clinical assessment is mandatory to better identify congestion in HF.

Reviewer 2 Report

I found the manuscript entitled: "Clinical, Laboratory and Lung ultrasound assessment of congestion in patients with Acute Heart Failure" very interesting. Lung ultrasound is important in the diagnosis and monitoring of the course of acute heart failure, hence the topic undertaken by the authors is of great clinical importance. The paper is methodologically well prepared and statistically elaborated, and the results obtained confirm the necessity of using this method in everyday practice. 
Minor revisions:
1. Chapter "Methods - lung ultrasound"  - last sentence - "inter- and intra-rater variability....." - note the values.
2. "Results" - all tables and figures should be numbered and signed.
3. Chapter "Predictors of outcome at admission" - the sentence "at multivariate analyses congestion score> 3 at admission........" is poorly formulated, incomprehensible.
4. Chapter "Discusssion" - in the first sentence - grammatical error, should be corrected to: "....at admission and discharge...." 

Author Response

Reviewer 2

I found the manuscript entitled: "Clinical, Laboratory and Lung ultrasound assessment of congestion in patients with Acute Heart Failure" very interesting. Lung ultrasound is important in the diagnosis and monitoring of the course of acute heart failure, hence the topic undertaken by the authors is of great clinical importance. The paper is methodologically well prepared and statistically elaborated, and the results obtained confirm the necessity of using this method in everyday practice. 

 We are grateful to this reviewer for his/her appreciation of our work and stat analysis

Minor revisions:
1. Chapter "Methods - lung ultrasound"  - last sentence - "inter- and intra-rater variability....." - note the values. Changed in “Inter- and intra-observer variability ranges from 5 to 8%”

  1. "Results" - all tables and figures should be numbered and signed. Figures and Tables are now labeled and described along the text
    3. Chapter "Predictors of outcome at admission" - the sentence "at multivariate analyses congestion score> 3 at admission........" is poorly formulated, incomprehensible.

Thanks for this correction we modified it in “ Multivariate analyses demonstrated congestion score>3 was an independent predictor of..
4. Chapter "Discusssion" - in the first sentence - grammatical error, should be corrected to: "....at admission and discharge...." done

Round 2

Reviewer 1 Report

The paper was improved and the authors have taken a serious step to improve the content. I have no additional meaningful comments.